# Imipenem Resistance Mediated by *bla*_OXA-913_ Gene in *Pseudomonas aeruginosa*

**DOI:** 10.3390/antibiotics10101188

**Published:** 2021-09-29

**Authors:** Dong-Chan Moon, Abraham Fikru Mechesso, Hee-Young Kang, Su-Jeong Kim, Ji-Hyun Choi, Hyun-Ju Song, Soon-Seek Yoon, Suk-Kyung Lim

**Affiliations:** Bacterial Disease Division, Animal and Plant Quarantine Agency, 177 Hyeksin 8-ro, Gimcheon-si 39660, Gyeongsangbuk-do, Korea; ansehdcks@korea.kr (D.-C.M.); abrahamf@korea.kr (A.F.M.); kanghy7734@korea.kr (H.-Y.K.); kimsujeong27@gmail.com (S.-J.K.); wlgus01@korea.kr (J.-H.C.); shj0211@korea.kr (H.-J.S.); yoonss24@korea.kr (S.-S.Y.)

**Keywords:** *bla*
_OXA-913_, imipenem, *P. aeruginosa*, resistance mechanism

## Abstract

Treatment of infectious diseases caused by carbapenem-resistant *Pseudomonas aeruginosa* is becoming a greater challenge. This study aimed to identify the imipenem resistance mechanism in *P. aeruginosa* isolated from a dog. Minimum Inhibitory Concentration (MIC) was determined by the broth microdilution method according to the Clinical and Laboratory Standards Institute recommendations. We performed polymerase chain reaction and whole-genome sequencing to detect carbapenem resistance genes. Genomic DNA of *P. aeruginosa* K19PSE24 was sequenced via the combined analysis of 20-kb PacBio SMRTbell and PacBio RS II. Peptide-Peptide Nucleic Acid conjugates (P-PNAs) targeting the translation initiation region of *bla*_OXA-913_ were synthesized. The isolate (K19PSE24) was resistant to imipenem and piperacillin/tazobactam yet was susceptible to most of the tested antimicrobials. Whole-genome sequencing revealed that the K19PSE24 genome comprised a single contig amounting to 6,815,777 base pairs, with 65 tRNA and 12 rRNA genes. K19PSE24 belonged to sequence type 313 and carried the genes *aph(3)-IIb*, *fosA*, *catB7*, *crpP*, and *bla*_OXA-913_ (an allele deposited in GenBank but not described in the literature). K19PSE24 also carried genes encoding for virulence factors (exoenzyme T, exotoxin A, and elastase B) that are associated with adhesion, invasion, and tissue lysis. Nevertheless, we did not detect any of the previously reported carbapenem resistance genes. This is the first report of the *bla*_OXA-913_ gene in imipenem-resistant *P. aeruginosa* in the literature. Notably, no viable colonies were found after co-treatment with imipenem (2 µg/mL) and either of the P-PNAs (12.5 µM or 25 µM). The imipenem resistance in K19PSE24 was primarily due to *bla*_OXA-913_ gene carriage.

## 1. Introduction

*Pseudomonas aeruginosa* is one of the most problematic opportunistic human pathogens and is particularly evident in cases of hospital-acquired pneumonia in immunocompromised patients. It can be transmitted from humans to companion animals or vice versa [1,2]. Carbapenems are currently the antibiotics of choice for the treatment of infections caused by multidrug-resistant pathogens. Carbapenems bind to penicillin-binding proteins and inactivate an inhibitor of autolytic enzymes within the cell wall which leads to the death of the bacteria [3]. Currently, different kinds of carbapenems are used in clinical practice as antipseudomonal agents, such as doripenem, imipenem, and meropenem [4]. Among these, meropenem is commonly used to treat bacteremia, sepsis, and infections caused by resistant bacteria in dogs and cats [5,6]. Adverse events are relatively rare with carbapenems and are mostly minor [7].

The global emergence of carbapenemase-producing bacteria is an alarming signal, potentially leading to ever-increasingly restricted therapeutic choices [8]. Carbapenemases are primarily classified into three classes of β-lactamases, the ambler classes A, B, and D β-lactamases. Class D β-lactamases, also known as oxacillinases (OXA), have become the most common type of acquired carbapenemases [9]. They are characterized by rapid mutation and an expanded spectrum of activity [10]. Carbapenem-hydrolyzing class D β-lactamases have been observed in *Enterobacterales* with OXA-48-like; in *Acinetobacter baumannii* with OXA-23-like, OXA-40-like, OXA-58-like, and OXA-143-like; and in *P*. *aeruginosa* with OXA-40-like, OXA-48-like, OXA-181-like, and OXA-198-like [10,11,12,13,14]. Recent emerging mechanisms of carbapenem resistance accumulate through the spread of carbapenem-destroying-β-lactamases and leave a narrowed range of therapeutic options. In this study, we aimed to determine the imipenem resistance mechanism in a clinical isolate of *P*. *aeruginosa* recovered from a dog with pyoderma.

## 2. Results and Discussion

PacBio SMART analysis demonstrated that the genome of *P. aeruginosa* (K19PSE24) comprised 6,815,777 base pairs with total coverage of 151.0× and 66.11% GC content. Abundant categories in the Cluster of Orthologues Groups (COG) distribution (>5% of the total COG-matched counts) include amino acid and inorganic ion transport and metabolism. We also identified virulence factors that are associated with focal adhesion, phagocytosis, and subsequent dissemination of *P. aeruginosa* (exoenzyme T, exoT), tissue lysis and invasion (exotoxin A, exoA), and acute infection (elastase B, lasB) [15,16]. Multilocus sequence typing demonstrated that the strain belonged to sequence type 313, a type already reported in patients from different countries, including South Korea [17,18,19].

In this study, *P. aeruginosa* (K19PSE24) was resistant to imipenem (MIC > 8 µg/mL) and piperacillin/ tazobactam (128/4 µg/mL), while it was susceptible to most of the tested antimicrobials, including meropenem and ceftazidime (Appendix A). K19PSE24’s chromosome possessed genes encoding resistance to aminoglycosides (*aph(3)-IIb*), β-lactams (*bla*_OXA-913_ and *bla*_PAO_), fosfomycin (*fosA*), phenicols (*catB7*), and quinolones (*crpP*). We did not, however, detect any of the previously reported carbapenem resistance-encoding genes using PCR and a whole-genome sequencing assay. Acquisition of multiple imported and chromosomally encoded resistance mechanisms and/or a single mutational event contribute to multidrug resistance in *P. aeruginosa* [20]. Fosfomycin, one of the oldest antimicrobials, has now been revisited for its possible effectiveness against multidrug-resistant strains, including *P. aeruginosa* [21]. Thus, further investigations may be needed on the contribution of the *fosA* gene on the susceptibility of *P. aeruginosa* to fosfomycin. Several studies have demonstrated the prevalence of the *bla*_PAO_ gene, a chromosomally-encoded cephalosporinase gene, in *P. aeruginosa* [22,23,24]. Madaha et al. [25] reported that *bla*_PAO_-carrying *P. aeruginosa* isolates were sensitive to carbapenems especially imipenem. The *bla*_OXA-913_ gene has also been reported in *P. aeruginosa* in Switzerland (GenBank: NG-068184.1); however, the finding has not been described in the literature. Here, we report for the first time the *bla*_OXA-913_ gene in imipenem-resistant *P. aeruginosa* isolated from a dog with pyoderma in South Korea. We found low similarity (36.5%) between the amino acid sequences of the *bla_O_*_XA-913_ gene detected in this study and the *bla*_OXA-48_-like gene in *K. pneumoniae* (GenBank accession number CP024838.1). In addition, except for a single amino acid substitution (Asn99Lys), *bla*_OXA-913_ in this study was highly similar (99.6%) to the *bla*_OXA-488_ gene identified in *P. aeruginosa* (NG-049768.1). The chromosomal regions of the *bla*_OXA-913_ gene had a substantial sequence homology (>98% sequence identity) compared to those of previously reported OXA-50 family class D genes in *P. aeruginosa* from different sources (Figure 1). Furthermore, we noted co-linearity (identity ≈ 99%) between the nucleotide sequences of our strain (CP053687) and those of the OXA-50 family gene-carrying *P. aeruginosa* strains described in Figure 1, as well as that of *P. aeruginosa* PAO1 (NC_002516.2). The detection of a new carbapenem resistance gene (*bla*_OXA-913_) in a virulent strain of *P. aeruginosa* is a great concern, since it significantly restricts the therapeutic options for patients.

PNA is an artificially synthesized DNA mimic that forms a stable complex with DNA and RNA molecules in a sequence-dependent manner. Previous studies have shown the target-specific gene silencing and/or growth inhibitory activities of P-PNAs in Gram-positive and Gram-negative bacteria [26,27]. Ray and Norden [28] revealed that P-PNAs can form a stable link with DNA and interfere with replication or transcription of the target genes. In this study, the P-PNAs were designed to target the translation initiation region of *bla*_OXA-913_. Growth inhibition was not observed when *P. aeruginosa* (K19PSE24) was treated with either the P-PNAs (0.4–25 µM/mL) or imipenem (2 µg/mL). However, no viable colonies were recovered after co-treatment with imipenem (2 µg/mL) and either 12.5 µM/mL or 25 µM/mL of the P-PNAs. Therefore, the imipenem resistance in K19PSE24 was primarily due to *bla*_OXA-913_ gene carriage.

## 3. Materials and Methods

### 3.1. Isolation and Identification of P. aeruginosa

An isolate of *P. aeruginosa* was obtained from a skin scraping specimen of a dog with pyoderma in 2019. Isolation and identification of *P. aeruginosa* were performed using a CHROMagarTM *Pseudomonas* agar plate (CHROMagar, Becton Dickinson, Sparks, MD, USA). The isolate was then confirmed by matrix-assisted laser desorption ionization-time-of-flight mass spectrometry (MALDI-TOF, bioMérieux, Marcy L’Etoile, France).

### 3.2. Antimicrobial Susceptibility Testing

Testing for antimicrobial susceptibility was performed by the broth microdilution method using the COMPGN1F Sensititre panel (Trek Diagnostic Systems, Cleveland, OH, USA), according to the manufacturer’s instruction. The results were interpreted according to the Clinical and Laboratory Standards Institute (CLSI) breakpoints. *P. aeruginosa* ATCC27853 was used as a reference strain.

### 3.3. Polymerase Chain Reaction (PCR) and Whole-Genome Sequencing

A PCR assay was performed to detect the most frequently reported carbapenem resistance genes in South Korea (*bla*_IMP_, *bla*_VIM_, *bla*_OXA-48_-like, *bla*_NDM_, and *bla*_KPC_) (Appendix A) [29,30]. Whole-genome sequencing was performed using PacBio RS II (Pacific Biosciences, Menlo Park, CA, USA), as previously described [31]. Antimicrobial resistance genes were analyzed by the Center for Genomic Epidemiology (http://www.genomicepidemiology.org/; accessed on 18 February 2021). The sequence type (ST) of *P. aeruginosa* (GenBank accession number CP053687.1) was inferred from *Pseudomonas* housekeeping genes using the Multilocus Sequence Typing Application 1.8 9 [32]. In addition, the sequence of the chromosomal regions of the *bla*_OXA-913_ gene in this study (CP053687) was compared with those of previously reported OXA-50 family class D genes. Briefly, the nucleotide sequences of complete genomes of the OXA-50 family class D gene-carrying *P. aeruginosa* were downloaded from the GenBank nucleotide database, sequences were trimmed and chromosomal regions (1Mb) containing *bla*_OXA_ genes were prepared. The average nucleotide identity (ANI) values were calculated with pairwise genome alignment of sequences by using the ANI-blast method implemented in PYANI (v.0.2.10) [33], and the phylogenetic tree was reconstructed based on the ANI values. In addition, the nucleotide sequence of our strain was compared with those of OXA-50 family class D gene-carrying strains used in ANI analysis, as well as that of *P. aeruginosa* PAO1 (NC_002516.2).

### 3.4. Peptide-Peptide Nucleic Acid Conjugation

We used artificially synthesized peptide-peptide nucleic acid conjugates (P-PNAs) ((KFF)3K-L-ATGCGCCCTCTCCTCTTCAG and (KFF)3K-L-CGAGCCATGCGCCCTCTCCT, 5′ to 3′ sequence) to silence the *bla*_OXA-913_ gene and confirm the gene associated with imipenem resistance. Briefly, the *bla*_OXA-913_-specific oligonucleotides were searched in the genome sequence of *P. aeruginosa* (CP053687.1). The resulting P-PNA oligomers for *bla*_OXA-913_ were designed to bind to its translation initiation region, which overlapped the ATG start codon and the ribosome-binding Shine-Dalgarno sequences (CGAGCC). PNA synthesis, purification, and conjugation with (KFF)3K-L bacterial penetration peptide were performed at PANAGENE (Daejeon, South Korea). *P. aeruginosa* (K19PSE24 and ATCC 27853, 5 × 10^4^ CFU/mL) were incubated with imipenem (2 µg/mL) alone or in combination with different concentrations of P-PNAs (0.4, 0.8, 1.6, 3.12, 6.25, 12.5, and 25 µM) in 100 µL of Muller-Hinton broth at 37 °C for four hours. Then, 50 µL was removed and spread-plated on Muller-Hinton agar plates, and the CFUs were determined after incubation at 37 °C for 18 h.

## 4. Conclusion

The detection of the *bla*_OXA-913_ gene in *P. aeruginosa* is an alarming emerging threat. This study highlights the need for continuous screening of companion animal isolates, given that a novel imipenem resistance gene was detected in an isolate recovered from a dog.

## Figures and Tables

**Figure 1 antibiotics-10-01188-f001:**
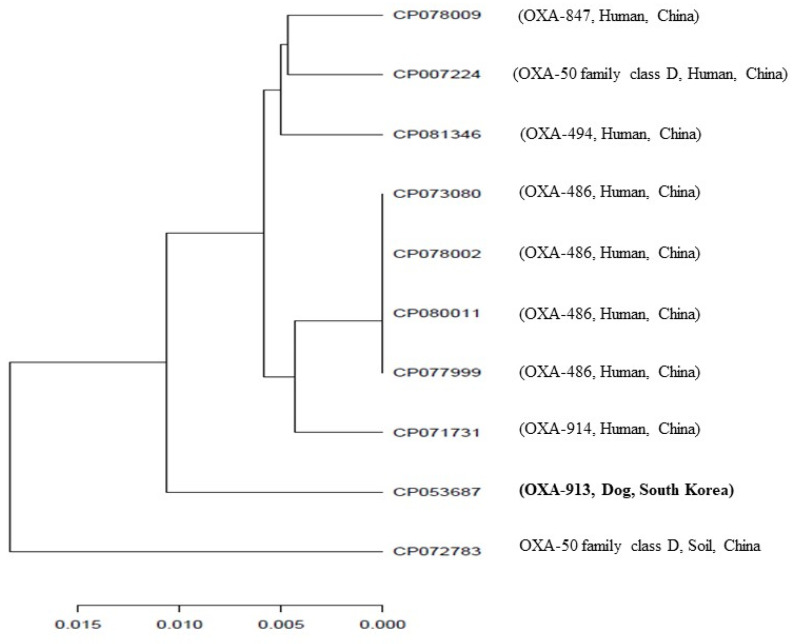
Comparative analysis of chromosomal regions (1 Mb) of the *bla*_OXA-913_ gene and OXA-50 family class D genes. ANI analysis was performed using the ANI-blast method implemented in PYANI (v.0.2.10) and the tree was generated based on the ANI values. The horizontal lines represent the 95% threshold value. The scale bar represents sequence divergence, i.e., the percentage of nucleotide substitution rate over the length of the genome.

## Data Availability

The nucleotide sequence of the K19PSE24 genome has been submitted to the GenBank nucleotide sequence database and assigned accession number CP053687.1.

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
