# Peer review of "Imipenem Resistance Mediated by blaOXA-913 Gene in Pseudomonas aeruginosa"

_antibiotics, 2021, doi:10.3390/antibiotics10101188_

Round 1
Reviewer 1 Report
In this manuscript entitled “Imipenem resistance mediated by blaOXA-913 gene in Pseudomonas aeruginosa”, the authors presented a genomic analysis of imipenem-resistant P. aeruginosa from a dog in South Korea. They isolated an imipenem-resistant P. aeruginosa from a dog with pyoderma and analyzed drug sensitivity and carbapenem-resistant genes. They also determined the whole-genome sequence of the isolate using a long-read sequencer and identified blaOXA-913 involved in carbapenem resistance. The manuscript is organized concisely. However, this research was of limited interest. The following comments should be considered.
- The authors identified blaOXA-913 as an imipenem-resistant gene. However, the clinical importance of OXA-913-producing P. aeruginosa was not discussed well.
- They also detected another beta-lactamase gene, blaPAO but did not discuss its contribution to beta-lactam resistance.
- Virulence genes were detected in the isolate?
- The phylogenetic analysis could be made to compare with other pathogenic P. aeruginosa genomes.
- 5. They identified other resistant genes, aph(3)-IIb, fosA, catB7 and crpP. However, they did not test fosfomycin and chloramphenicol sensitivity and nor detect aminoglycoside and quinolone resistance.
Author Response
Dear reviewer
We are grateful for the constructive comments and suggestions on our manuscript. We believe that your comments and suggestions are essential to improve the quality of our manuscript and therefore, we have modified it accordingly. Our responses to the raised questions/comments are summarized below.
Sincerely,
Suk-Kyung Lim (DVM, Ph.D.)
Point 1. The authors identified blaOXA-913 as an imipenem-resistant gene. However, the clinical importance of OXA-913-producing P. aeruginosa was not discussed well.
Response: Thank you for the comment. The detection of blaOXA-like gens in P. aeruginosa is commonly associated with β-lactam resistance, which subsequently leads to treatment failure in infected patients. Based on this a brief remark is added in lines 78-80 as follows "The detection of new carbapenem resistance gene (blaOXA-913) in P. aeruginosa is a great concern since it significantly restricts the therapeutic option for patients".
Point 2. They also detected another beta-lactamase gene, blaPAO but did not discuss its contribution to beta-lactam resistance.
Response: Thank you for your comment. We have included an explanation of the blaPAO gene and its implications in lines 72-75."Several studies have demonstrated the prevalence of blaPAO gene, a chromosomally-encoded cephalosporinase gene, in P. aeruginosa. Madaha et al. (21) reported that blaPAO carrying P. aeruginosa isolates were sensitive to carbapenems especially imipenem". In this study, our PNA was specifically designed to target the translation initiation region of blaOXa-913 gene. This implies that imipenem resistance in P. aeruginosa is primarily due to this blaOXa-913 gene carriage (lines 89-91).
Point 3. Virulence genes were detected in the isolate?
Response: We appreciate the comment and corrections have been made in lines 68-70. " We also identified virulence factors that are associated with focal adhesion, phagocytosis, and subsequent dissemination of Pseudomonas aeruginosa (exoenzyme T, exoT), tissue lysis and invasion (exotoxin A, exoA), and acute infection (elastase B, lasB)"
Point 4. The phylogenetic analysis could be made to compare with other pathogenic P. aeruginosa genomes.
Response: Agreeing with you, performing phylogenetic analysis would be better to demonstrate the similarity with other pathogenic strains. In the current study, however, our primary objective was to report the first detection of the blaOXA-913 gene in an imipenem-resistant isolate. We will consider the analysis in our subsequent studies.
Point 5. They identified other resistant genes, aph(3)-IIb, fosA, catB7, and crpP. However, they did not test fosfomycin and chloramphenicol sensitivity and nor detect aminoglycoside and quinolone resistance.
Response: We did a susceptibility study on additional antimicrobials (check the table below) but we provided only the results of those antimicrobials best recommended for Pseudomonas or with CLSI breakpoints.
Antimicrobial agent |
MIC (µg/mL) |
Amoxicillin/ Clavulanic Acid |
>32 |
Ampicillin |
>16 |
Cefazolin |
>16 |
Cefovecin |
>8 |
Cefoxitin |
>16 |
Cefpodoxime |
>16 |
Ceftiofur |
>4 |
Chloramphenicol |
16 |
Clindamycin |
>4 |
Doxycycline |
8 |
Enrofloxacin |
0.5 |
Marbofloxacin |
0.5 |
Rifampin |
>2 |
Ticarcillin |
64 |
Trimethoprim/ Sulfamethoxazole |
>2 |
Reviewer 2 Report
The manuscript is definetly of interest to be published and should be accepted on the condition that the following concerns are addressed:
General:
the manuscript is well-written, uses concise and scientific language
only minor English changes and corrections are needed
Introduction
L30: please include the following reference:
https://pubmed.ncbi.nlm.nih.gov/33406652/
L32: please include the following referece:
https://academic.oup.com/cid/article/43/Supplement_2/S49/332756
Please discuss in more detail and provide references that carbapenems are often the last safe and effective alternatives to treat infections, as others (e.g. colistin, tigecycline) may have severe adverse events.
L44-45: the research idea is introduced rather abruptly. please include some more sentences.
Discussion:
please discuss the relevance and possibilities of carbapenem-resistant but cephalosporin susceptible Pseudomonas
Author Response
Dear reviewers
We are grateful for the constructive comments and suggestions on our manuscript. We believe that your comments and suggestions are essential to improve the quality of our manuscript and therefore, we have modified it accordingly. Our responses to the raised questions/comments are summarized below.
Sincerely,
Suk-Kyung Lim (DVM, Ph.D.)
Point 1. L30: please include the following reference: https://pubmed.ncbi.nlm.nih.gov/33406652/
Response: Thank you for the suggestion but we believe that reference 1 is sufficient enough for this section
Point 2. L32: please include the following reference: https://academic.oup.com/cid/article/43/Supplement_2/S49/332756
Response: Thank you for the suggestion and the reference is included accordingly (line 41).
Point 3. Please discuss in more detail and provide references that carbapenems are often the last safe and effective alternatives to treat infections, as others (e.g. colistin, tigecycline) may have severe adverse events.
Response: The following general information on carbapenems are included in lines 33-39." Carbapenems bind to penicillin-binding proteins and inactivate an inhibitor of autolytic enzymes within the cell wall which lead to the killing of the bacteria [2]. Currently, different kinds of carbapenems are used in clinical practice as antipseudomonal agents such as doripenem, imipenem, and meropenem [3]. Among these, meropenem is commonly used to treat bacteremia, sepsis, and infections caused by resistant bacteria in dogs and cats [4,5]. Adverse events are relatively rare with carbapenems and are mostly minor [6].
Point 4. L44-45: The research idea is introduced rather abruptly. please include some more sentences.
Response: Thank you for the comment and the following statement is added: "Recent emerging mechanisms of resistance accumulate through the spread of carbapenem-destroying-lactamases leaving narrow therapeutic options". (Lines 48-50)
Discussion:
Point 5: Please discuss the relevance and possibilities of carbapenem-resistant but cephalosporin susceptible Pseudomonas
Response: Our isolates exhibited high MIC to most of the tested cephalosporins (Table below). Thus, the discussion for "the possibility of carbapenem-resistant but cephalosporin susceptible Pseudomonas" in this case may not be important.
Antimicrobial agent |
MIC (µg/mL) |
Cefazolin |
>16 |
Cefovecin |
>8 |
Cefoxitin |
>16 |
Cefpodoxime |
>16 |
Ceftiofur |
>4 |
Ceftazidime |
≥32 |
Reviewer 3 Report
The authors present an intriguing short communication about the mechanism responsible for imipenem resistance in a canine strain of Pseudomonas. While the short nature of the report is acceptable for the publication mechanism, the authors must show data in the main text of the article beyond the MIC data in the supplementary section. My suggestion would be that the authors add figures for the knock-down of blaOXA-913 gene expression using PNA (Western or qPCR), and some images of the bacterial plating data to support their conclusions. While this will increase the size of the paper, and possibly change the article type, I believe that these suggested changes will enhance the readability of the article.
Author Response
Dear reviewer
We are grateful for the constructive comments and suggestions on our manuscript. We believe that your comments and suggestions are essential to improve the quality of our manuscript and therefore, we have modified it accordingly. Our responses to the raised questions/comments are summarized below.
Sincerely,
Suk-Kyung Lim (DVM, Ph.D.)
Point: The authors present an intriguing short communication about the mechanism responsible for imipenem resistance in a canine strain of Pseudomonas. While the short nature of the report is acceptable for the publication mechanism, the authors must show data in the main text of the article beyond the MIC data in the supplementary section. My suggestion would be that the authors add figures for the knock-down of blaOXA-913 gene expression using PNA (Western or qPCR), and some images of the bacterial plating data to support their conclusions. While this will increase the size of the paper, and possibly change the article type, I believe that these suggested changes will enhance the readability of the article.
Response: Thank you for the constructive comments. Western blotting or qPCR data would have vital importance to further confirm our findings. However, we did not have the data of western blotting or qPCR for this specific case. We preferred to use PNA to silence the blaOXA-913 gene and determine the mechanism of imipenem resistance. We have the images of the bacterial plating data (bacteria treated with imipenem alone and in combination with imipenem). However, we believe that adding the images (plates with bacterial growth or not) may not have relevance to the manuscript because the findings are clearly explained in lines 84-87.
Round 2
Reviewer 1 Report
Point 5. They identified other resistant genes, aph(3)-IIb, fosA, catB7, and crpP. However, they did not test fosfomycin and chloramphenicol sensitivity and nor detect aminoglycoside and quinolone resistance.
Response: We did a susceptibility study on additional antimicrobials (check the table below) but we provided only the results of those antimicrobials best recommended for Pseudomonas or with CLSI breakpoints.
Please add a cautious description that the authors did not confirm the contribution of fosfomycin, chloramphenicol, aminoglycoside, and quinolone resistance genes (aph(3)-IIb, fosA, catB7, and crpP).
Author Response
Dear reviewer,
We appreciate the positive comment on our manuscript. Please find the response below to your comments or suggestions.
Sincerely,
Point: Please add a cautious description that the authors did not confirm the contribution of fosfomycin, chloramphenicol, aminoglycoside, and quinolone resistance genes (aph(3)-IIb, fosA, catB7, and crpP).
Response: Low permeability of the outer membrane and efflux mechanism(s) are likely to contribute significantly to the intrinsic resistance of P. aeruginosa isolates to tetracycline, chloramphenicol, and fluoroquinolones (Li et al., 1994; Antimicrobial agents and chemotherapy 38, 1732-1741). In addition, resistance to aminoglycosides with antipseudomonal activities, including gentamicin, tobramycin, and amikacin is also too common and is present virtually throughout the world (Poole 2005, Antimicrobial agents and chemotherapy 49, 479-487). Therefore, considering the objective of this study, susceptibility studies on P. aeruginosa to these antimicrobials will have minor significance. Fosfomycin, one of the oldest antimicrobials, has now been revisited for its possible effectiveness against multidrug-resistant strains including P. aeruginosa (De Groote et al., 2011 Journal of Medical Microbiology 2011, 60, 329-336). Thus, further investigations may be needed on the contribution of fosA gene on the susceptibility of P. aeuroginsa to fosfomycin. (Lines 81-84)
Reviewer 3 Report
I would like to thank the authors for their comments. Unfortunately, I do not think that the manuscript meets the threshold of a minimal publishable unit without the presentation of additional data (not a description of data). Given the importance of the question addressed, I hope that the authors will add something other than (in addition to) the MIC chart.
Author Response
Point: I would like to thank the authors for their comments. Unfortunately, I do not think that the manuscript meets the threshold of a minimal publishable unit without the presentation of additional data (not a description of data). Given the importance of the question addressed, I hope that the authors will add something other than (in addition to) the MIC chart.
Response
Dear reviewer,
We appreciate your concern. As we have explained in our previous response, data on western blotting or qPCR would have vital importance to explain the contribution of blaOXA-913 gene. However, we believe that the MIC, WGS, and peptide-nucleic acid conjugates (P-PNAs) experiments are also sufficient to confirm the association between imipenem resistance and blaOXA-913 gene in P. aeruginosa. Recently, several researchers are studying the antibacterial activity P-PNAs against multidrug-resistant bacteria. Therefore, we believe that our data is sufficient for this manuscript, short communication.